# The effectiveness of savouring interventions on well-being in adult clinical populations: A protocol for a systematic review

Katie Cullen[1]*, Mike Murphy[1], Zelda Di Blasi[1], Fred B. Bryant[2]

1 School of Applied Psychology, University College Cork, Cork, Ireland, 2 Department of Psychology, Loyola University of Chicago, Chicago, Illinois, United States of America

☯ These authors contributed equally to this work.
* 121101313@umail.ucc.ie

## Abstract

### Purpose

Savouring interventions aim to amplify the intensity and duration of positive feelings and positive affect. Research has shown that the potential benefits of savouring include the promotion of psychological well-being and diminution of negative affective states. Savouring strategies may be particularly useful amongst clinical populations in changing biobehavioural processes which can strengthen an individual's propensity to exert control over how to develop, intensify and promote psychological well-being, while simultaneously mitigating negative affective states. This paper outlines a protocol for a systematic review that will be used to identify, critically appraise and synthesise findings of studies examining the effectiveness of savouring interventions in adult clinical populations. Savouring interventions will be defined broadly, operationalised as any savouring strategy focusing on past, present or future events or experiences whereby participants are instructed to attend to and amplify positive affect relating to same. The goal of our review is to include the extent of the literature on this topic and contribute to the overall evidence to support savouring interventions.

### Methods

This protocol is carried out in accordance with the Preferred Reporting Items for Systematic Reviews and Meta-Analyses (PRISMA) Protocols Guidelines. The protocol has been registered with PROSPERO (CRD42023404857). The databases PsycINFO, PubMed, CINAHL (Cumulative Index to Nursing and Allied Health Literature), and Scopus will be searched alongside a search of grey literature. An examination of the first 200 papers on Google Scholar will also be done to identify relevant papers. Studies describing randomised controlled trials evaluating the effects of savouring interventions as described within this article on adult clinical populations will be included in the review. Outcomes will include well-being, quality of life, depression, anxiety or stress. Study selection and data extraction will be completed by three independent reviewers to reduce risk of bias. Interrater percentage agreement and interrater reliability will be reported on same. The quality of studies will be assessed using criteria based on the Cochrane Collaboration's tool for assessing risk of

**Data Availability Statement:** No datasets were generated or analysed during the current study. All relevant data from this study will be made available upon study completion.

**Funding:** The author(s) received no specific funding for this work.

**Competing interests:** The authors have declared that no competing interests exist.

bias and the Jadad scale. A narrative synthesis with tables of study characteristics will be provided. Separate considerations of the three time perspectives of savouring interventions (past-focused, present-focused, and future-focused) will also be described.

## Conclusions

This systematic review will provide important clinical insights into the potential efficacy of savouring interventions when working with adult clinical samples.

## Introduction

Savouring has been described by Bryant [1] as a set of behavioural and/or cognitive strategies that aim to amplify the intensity and duration of positive feelings and positive affect. It has been proposed as a regulatory process which influences the relationship between positive events plus an individual's positive emotional reactions to these events [2]. This term was coined with a view of capturing the active process by which individuals perceive and experience enjoyment, referencing an ongoing interplay between a person and their environment [3]. Savouring may be classified as a positive psychology intervention, as it pertains to processes that aim to regulate, intensify as well as strengthen positive emotions and experiences [4]. However, savouring is an unique concept which is distinguishable from a variety of related but separate constructs, such as mindfulness, meditation, and generating flow [5]. Bryant and Veroff [5] highlight that an ability to be mindful, use focal attention and engage in meta-awareness is necessary to actively attempt to amplify positive emotions associated with positive experiences, events and memories. They describe the process of savouring as including a series of cognitive and regulatory responses where one firstly notices and attends to something positive, and subsequently responds cognitively or behaviourally to this stimulus, experiencing positive emotional reactions as a consequence [5,6].

Savouring strategies have been implemented into interventions within the positive psychology literature in numerous ways [7]. Examples of savouring strategies proposed by Bryant and Veroff [3] include sharing positive experiences with others, positive memory building, behavioural expression (laughing or showing affect), counting blessings, self-congratulation (regulating pride), engaging in positive imagination or fantasizing (e.g. attaining goals, imagining one's "Best Possible Self"), reminiscing on past positive events and engaging in active-constructive communication with others [5]. Personal savouring interventions encourage an individual to reflect on personal positive events, memories or experiences, whereas relational savouring interventions consider positive moments one has had with other individuals, such as children, spouses, or other prominent individuals [8]. Other savouring interventions include engaging in intentional positive activities such as practising kindness and expressing gratitude [9]. Interventions which include engagement with savouring strategies encourage the process of reliving, anticipating or attending to positive feelings associated with an experience or an event, whether real or imagined [10,11]. Research has shown that the potential benefits of savouring include the promotion of psychological well-being in addition to a decrease in experiencing negative affective states, most notably within depressive psychopathology [12,13]. Additionally, savouring processes can contribute to the development and maintenance of positive relationships with others, impact positively on mental and physical health outcomes, and can contribute to psychological agency as one ages [5,13,14].

Savouring interventions encompass three timeframes [10]. These include interventions which emphasise the development of savouring future positive events before they occur

(anticipation), present positive events while they are unfolding (savouring the present moment), or past positive events after they occur (reminiscence) [15]. These interventions aim to increase the well-being of an individual through the PERMA pathway (Positive Emotion, Engagement, Relationships, Meaning and Accomplishment) of savouring pleasant moments or experiences [16] as proposed by Seligman [17]. Different savouring processes regulate different positive emotional states [18]. Savouring processes include thanksgiving (regulating gratitude), marvelling (regulating awe), basking (regulating pride), and luxuriating (regulating pleasure) [10]. These interventions may be applied in singularity or as part of a multicomponent programme.

Bryant and Veroff's early work [19,20] highlighted a key distinction between an individual's ability to cope with negative experiences and their capacity to derive pleasure from positive experiences. They proposed that just as individuals utilize a set of cognitive and behavioural strategies to assist them in coping with processing negative events and regulate negative emotions, separate mechanisms exist for the processing of positive events and regulating of positive emotions. Research demonstrates that emotional dysregulation contributes greatly to the development and maintenance of a plethora of mental health difficulties including anxiety disorders, depressive disorders, eating disorders, bipolar affective disorder and personality disorders [21–25]. Current evidence now suggests that savouring may be effective in changing biobehavioural processes which can strengthen an individual's ability to derive pleasure from positive experiences [15,26]. It's proposed that the practise of upregulating positive emotions could facilitate well-being while also mitigating symptoms of psychological distress [27]. Individuals living with chronic mental health or life limiting physical health conditions may be more likely to experience lower levels of positive affect or transiently positive emotional experiences [28], which may further dispose them to future psychological distress [29,30]. This indicates that an individual's disposition to initially experience positive affect states, and secondly attend to and savour them, may be impacted by their psychological and physical levels of functioning and well-being [31,32]. Consequently, there is a need to examine the efficacy of savouring interventions when applied exclusively to clinical populations, as their therapeutic potential on altering positive and negative affect states may operate through different pathways than when applied to healthy, non-clinical populations [33,34]. The goal of our review is to include the extent of the literature on this topic and contribute to the overall evidence to support savouring interventions.

Systematic reviews have previously been published which highlight the efficacy of particular savouring processes in improving participant levels of well-being in addition to reducing symptoms of psychological distress, for example when practising gratitude [35] or when imagining one's "Best Possible Self" [36]. However to date, no systematic review has been published which reports the overall effectiveness of savouring interventions when utilising a broad definition of such interventions as is being proposed in the current study, and is expanded on in the methodology section of this paper. We believe that utilising a broad definition of a savouring intervention fits well with the growing size, disciplinary reach and impact of the research being conducted in this area [37]. Therefore, this proposal outlines our plan conduct a systematic review which aims to evaluate the efficacy of savouring interventions on well-being, quality of life, depression, anxiety or stress amongst adult clinical samples. This systematic review will synthesise the literature to report on same, and report a narrative synthesis with comparisons across past-, present-, and future-focused savouring interventions.

## Methods

The updated 2020 Preferred Reporting Items for Systematic Reviews and Meta-Analyses (PRISMA) guidelines [38] (outlined in S1 Checklist) and recommendations from the

Cochrane Review Group [39] informed the planning and the implementation of this systematic review protocol. The study protocol was registered with PROSPERO and the registration number is CRD42023404857.

The PICO Framework (population, intervention, comparison and outcomes) [40] was utilised to formulate the systematic review's research questions and to ensure that relevant components of the research question were well defined.

The review questions are:

1. How effective are savouring interventions when applied to adult clinical samples on well-being, quality of life, depression, anxiety or stress, when compared with comparison control groups?

2. What is the methodological quality of the evidence base for the effects of these savouring interventions, based on items from the Cochrane Collaboration's tool for assessing risk of bias and the Jadad scale?

## Inclusion criteria

Table 1 outlines eligibility criteria for inclusion in the review. Papers will be included in the review if they are written in English, published in peer-reviewed journals or grey literature, describe savouring interventions as outlined below, utilize a randomized-controlled design (randomized controlled trials [RCTs], cluster RCTs, or trials involving matched pairs with randomization). Randomization is a fundamental inclusion criterion as this design feature greatly reduces risk of bias [41,42]. The decision to only include studies published in English in our review was made when considering the limited resources and the language constraints of our

**Table 1. Inclusion criteria.**

| Inclusion Criteria | Present | Absent |
|---|---|---|
| **Design:**<br> Randomised Controlled Trial | | |
| **Participants:**<br> Age ($\geq$ 18 years)<br> Clinical population (mental or physical health diagnosis given by medical professionals and/or based on meeting significant cut off points on validated psychometric instruments) | | |
| **Intervention:**<br> Savouring interventions. These include:<br> • Interventions which instruct participants to attempt to attend to, intensify and prolong positive emotions or positive affect associated with a positive stimulus, whether that be real or imagined.<br> • May emphasise the development of savouring future positive events before they occur (anticipation), present positive events while they are unfolding (savouring the present moment), or past positive events after they occur (reminiscence).<br> • Must instruct participants to attend mindfully, reminisce, review, anticipate positive events, experiences or memories only (not integrate meaning from negative or neutral events).<br> • Examples include: thinking, talking or writing about positive events, counting blessings, noticing 'Three Good Things', reflecting on kindness (done to and performed by the individual) or engaging in any specific strategy to increase positive emotions attached to an event or the present moment e.g. implementing temporal scarcity into everyday experiences or enhancing active-constructive communications with others. | | |
| **Comparisons:**<br> Waiting list control group or alternative intervention control group (e.g. treatment as usual, placebo control groups, or another well-defined treatment such as cognitive behavioural therapy) | | |
| **Outcome measures:**<br> Validated measures of well-being (happiness, life satisfaction), quality of life, depression, anxiety or stress | | |

review team. Including studies published in languages other than English would have greatly challenged the review team in terms of sourcing expertise in non-English languages, cost and time. There will be no limitations placed on the professional background of people facilitating savouring interventions, the settings in which interventions are delivered, or the mode of delivery.

Papers must also meet the additional criteria outlined below to be included in the review:

*Design*: randomise controlled trials evaluating the efficacy of savouring base interventions will be included.

*Participants*: adults ($\geq$ 18 years) of any gender, who present with clinical symptoms or disorders. Participant mental health or physical health diagnosis must be given by medical professionals and/or based on meeting significant cut off points on validated psychometric instruments.

*Intervention*: papers must describe savouring interventions. These include interventions which instruct participants to attempt to attend to, intensify and prolong positive emotions or positive affect associated with a positive stimulus, whether real or imagined. This may include emphasising the development of savouring future positive events before they occur (anticipation), savouring present positive events while they are unfolding (savouring the present moment), or savouring past positive events after they occur (reminiscence). Examples include thinking, talking or writing about positive events, counting one's blessings, noticing 'Three Good Things', reflecting on kindness (done to and performed by the individual) or engaging in any specific strategy to increase positive emotions attached to an event or the present moment e.g. adopting a positive attentional focus, engaging in sensory-perceptual sharpening, implementing temporal scarcity into everyday experiences or enhancing active-constructive communications with others.

*Comparisons*: studies with a waiting control group or alternative intervention control groups (e.g. treatment as usual, placebo control groups, or another well-defined treatment such as cognitive behavioural therapy) will be included.

*Outcome measures*: papers must include a primary outcome measure of well-being (e.g. happiness, life satisfaction), quality of life, depression, anxiety or stress. Assessment of intervention and control group outcomes must be assessed by self-report, clinician or proxy administered and validated psychometric instruments. Outcome measures may include but are not limited to the following tools; the Life Satisfaction Index, the Quality of Life Scale, The Center for Epidemiological Studies Depression Scale, the Generalized Anxiety Disorder 7-item Scale and the Perceived Stress Scale [43–47].

## Exclusion criteria

*Design*: Papers will be excluded if they report exclusively qualitative studies, uncontrolled studies, or non-randomized trials. Discursive, theoretical, and narrative review papers, editorials, letters, commentaries, systematic reviews and meta-analyses will be excluded.

*Participants*: Papers describing trials of savouring interventions delivered to children, adolescents or those not diagnosed with a physical or mental health condition will be excluded. Papers examining the effectiveness of savouring interventions on healthy adults who include outcomes such as well-being, quality of life, depression, anxiety or stress will not be included.

*Intervention*: Interventions which instruct participants to engage in techniques such as reminiscence, life review or mindful attention which do not restrict participants' focus to solely positive events, positive stimuli or positive memories will be excluded. Emotional regulation training interventions which do not specifically instruct the development of positive emotions, feelings or affect will be excluded. Multi-component interventions which include a savouring activity or technique as part of a broader multi-element component will also not be included.

**Table 2. PsycINFO search terms for systematic review.**

*AB (savoring OR savouring OR "positive reminisc*" OR basking OR "positive life review" OR "behavioural expression" OR "memory building" OR "self-congratulation" OR "positive anticipation" OR "positive emotion regulation" OR appreciation OR "mindful attention" OR awe OR wonder OR luxuriat* OR bask* OR thanksgiving OR marvel* OR "counting blessings" OR "three good things")*

*Comparison*: papers evaluating savouring interventions with no comparison control groups will be excluded.

*Outcome measures*: RCTs of savouring interventions which do not include an outcome of well-being, quality of life, depression, anxiety or stress will be excluded. Assessment of intervention and control group outcomes must be assessed by self-report, clinician or proxy administered and validated psychometric instruments.

All criteria must be met for a paper to be eligible:

## Identification and selection of studies

**Search strategy.** The following electronic databases will be searched: PsycINFO, PubMed, CINAHL (Cumulative Index to Nursing and Allied Health Literature), and Scopus. Searches will be performed on databases from inception to present. No limitations will be imposed on date of publication. The same search strategy will be used for each database, but with alterations made to accommodate database interface. Consultation with a qualified librarian experienced in conducting systematic reviews was sought when drafting and reviewing search strategies for each database before searches were run, in order to minimize bias. The different terms identified to be used in searches can be seen in Table 2, with search strategies, with specific terms used, including Boolean operators for each database available in S1 File.

Additional manual searches will be conducted in which 1) table of contents of key journals will be searched, and 2) reference lists of previous relevant systematic reviews and meta-analysis will be searched, including reference lists of included and excluded primary studies and 3) forward and backward citation searches will be completed for all included papers.

A grey literature search will also be carried out to include studies not published in peer-reviewed journals [48] (*http://www.greynet.org/greysourceindex.html*). The first 200 results, sorted by relevance, following a search on Google Scholar will also be searched for primary studies meeting the inclusion criteria outlined above [49].

**Study screening and selection.** Retrieved records will be uploaded to Endnote (http://endnote.com) where duplicates will be removed and then imported into Rayyan software (https://www.rayyan.ai/). The primary researcher (KC) will screen records by title and abstract and by full text. Two independent reviewers (MM and ZD) will screen 10% of records independently, as recommended by the Cochrane Review Group to ensure screening efficiency [50,51]. Disagreements between reviewers about screening and selection will be resolved by discussion and consensus, and inter-rater reliability and percentage agreement will be computed for same. Reference lists to included records and excluded full texts, sorted by reason for exclusion, will be contained in supplementary materials.

**Inter-rater agreement and reliability of codin.** KC, MM, ZBD and FBB contributed to the development of a screening tool outlining the reviews inclusion and exclusion criteria, which is available in S2 File. Raters will familiarise themselves with this screening tool prior to screening articles for inclusion in the review in order to ensure reliability of coding across individual raters. KC, MM and ZBD will contribute to screening articles at both title and abstract and full text stages. Decisions of inclusion and exclusion of primary studies will be noted by individual raters, and disagreements resolved through discussion until a consensus is reached.

In the event a consensus cannot be reached, consultation with FBB will occur. Inter-rater reliability will be reported using percentage agreement and Krippendorff's alpha (K-alpha) [52] statistics.

## Assessment of characteristics of studies

**Data extraction.** A data extraction table created by the primary researcher will be used to extract data from included studies. Data extraction on 10% of studies will be completed by two pairs of raters (KC & MM, KC & ZD) until a Krippendorff's alpha (K-alpha) [52] value of .80 is achieved as recommended by Cohen [53]. KC will then complete data extraction of the remaining papers independently. Extracted information will include publication date and type (published or grey literature), country where the study was conducted, population characteristics (including age, gender, diagnosis/symptoms, recruitment method, and sample size per condition); intervention characteristics (including type of savouring intervention, delivery mode, number of sessions, duration in weeks), attrition rates, methodological characteristics (including study design, comparison group type), outcome assessment points (i.e., pre, post or follow up) and details of assessment instruments.

**Study quality assessment.** The quality of each primary study will be assessed using a seven-item scale, with criteria based on the Cochrane Collaboration's tool for assessing risk of bias [54] and the Jadad scale [55]. This rating scale has previously been used in other large-scale meta-analyses evaluating positive psychology interventions [56,57] and provides the research team with a checklist of seven items to assess the quality of each study. Items are rated as 0 if a criterion is absent or 1 if it is present. Studies will be classified as "good" if all seven criteria are met, "fair" if 5–6 criteria are met, or "poor" if 1–4 criteria are met.

**Data synthesis.** This is a review and synthesis of quantitative evidence. A narrative synthesis will be completed so that an exploration of the similarities and differences between studies can be highlighted, and so that an assessment of the strength of the evidence base evaluating savouring-based interventions can be conducted. Additionally, separate considerations of the three time perspectives of savouring interventions (past-focused, present-focused, and future-focused) will also be described. Tables of study characteristics will be provided.

**Ethics and dissemination.** The review has been registered with Prospero. The final article depicting the systematic review and its findings will be written in journal article format, and submitted to a relevant empirical journal for publication. Findings of the review will also be written up for the purpose of a Major Research Project as part of the University College Cork Doctorate in Clinical Psychology (DClinPsy).

**The status and timeline of the study.** The review is ongoing. We expect to complete it and report results in three months, by April 2024.

## Conclusions

The evidence drawn from this systematic review will contribute to our understanding of the clinical applications of savouring interventions when working with adult clinical samples. The findings from this review may add to the evidence base supporting these interventions and facilitate a greater understanding of factors which may contribute to the success or failure of their implementation.

## Supporting information

**S1 Checklist. PRISMA-P (Preferred Reporting Items for Systematic review and Meta-Analysis Protocols) 2015 checklist: Recommended items to address in a systematic review**

**protocol\*.**
(DOCX)

**S1 File.**
(DOCX)

**S2 File.**
(DOCX)

## Acknowledgments

We would like to express our thanks to the Academic Editor and two anonymous reviewers who provided our team with feedback and suggestions on how to improve the quality of our protocol.

## Author Contributions

**Conceptualization:** Katie Cullen, Mike Murphy, Zelda Di Blasi.

**Data curation:** Katie Cullen, Mike Murphy, Zelda Di Blasi.

**Formal analysis:** Katie Cullen, Mike Murphy, Zelda Di Blasi.

**Investigation:** Katie Cullen, Mike Murphy, Zelda Di Blasi, Fred B. Bryant.

**Methodology:** Katie Cullen, Mike Murphy, Zelda Di Blasi, Fred B. Bryant.

**Project administration:** Katie Cullen, Mike Murphy, Zelda Di Blasi.

**Resources:** Katie Cullen, Mike Murphy, Zelda Di Blasi.

**Software:** Katie Cullen.

**Supervision:** Mike Murphy, Zelda Di Blasi, Fred B. Bryant.

**Writing – original draft:** Katie Cullen.

**Writing – review & editing:** Katie Cullen, Mike Murphy, Zelda Di Blasi, Fred B. Bryant.

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
