## [Decision Letter · Decision Letter 0]

9 Jan 2024

PONE-D-23-36445The Effectiveness of Savouring Interventions on Well-being in Adult Clinical Populations: A Protocol for a Systematic ReviewPLOS ONE

Dear Dr. Cullen,

Thank you for submitting your manuscript to PLOS ONE. After careful consideration, we feel that it has merit but does not fully meet PLOS ONE’s publication criteria as it currently stands. Therefore, we invite you to submit a revised version of the manuscript that addresses the points raised during the review process. More detail is required regarding the literature search and data synthesis methods. The importance of reproducibility and replicability for systematic reviews cannot be overemphasized.

We look forward to receiving your revised manuscript.

Kind regards,

Qin Xiang Ng, MBBS, MPH

Academic Editor

PLOS ONE

Journal Requirements:

- https://doi.org/10.1080/17439760.2020.1818807

- https://bmcpsychiatry.biomedcentral.com/articles/10.1186/s12888-018-1739-2

In your revision ensure you cite all your sources (including your own works), and quote or rephrase any duplicated text outside the methods section. Further consideration is dependent on these concerns being addressed.

Additional Editor Comments:

1. It would be better to have two subheadings under the Methods section, e.g. “Identification and selection of studies” and “Assessment of characteristics of studies”.

2. Inclusion criteria should be spelled out as Design, Participants (not clear), Intervention (not clear, need to tidy up), Comparison (if applicable), Outcome measures.

3. Please provide the exact search terms and operators used for at least one database in the main manuscript, with the remaining details available in supplementary materials. For each database, it should be as example like: Embase (1947 to June 2023). This is missing.

4. Authors should tidy up information, perhaps using a table to cover all area, such as Design, Participants, Intervention, Outcome measures, Comparisons.

Reviewers' comments:

Reviewer's Responses to Questions

**Comments to the Author**

1. Does the manuscript provide a valid rationale for the proposed study, with clearly identified and justified research questions?

Reviewer #1: Yes

Reviewer #2: Partly

2. Is the protocol technically sound and planned in a manner that will lead to a meaningful outcome and allow testing the stated hypotheses?

Reviewer #1: Yes

Reviewer #2: Partly

3. Is the methodology feasible and described in sufficient detail to allow the work to be replicable?

Reviewer #1: Yes

Reviewer #2: No

4. Have the authors described where all data underlying the findings will be made available when the study is complete?

Reviewer #1: No

Reviewer #2: No

5. Is the manuscript presented in an intelligible fashion and written in standard English?

Reviewer #1: Yes

Reviewer #2: Yes

6. Review Comments to the Author

You may also provide optional suggestions and comments to authors that they might find helpful in planning their study.

Reviewer #1: Summary:

The authors have attempted to highlight the importance of savouring interventions on well being in adult clinical populations. Their main objective is to synthesise the literature and report a narrative synthesis with comparisons across the past, present and future focused savouring interventions.

Abstract:

Kindly rephrase “no systematic review to date has been published evaluating their efficacy with adult clinical populations.” This is misleading as there are SRs evaluating their efficacy in adult populations.

This is an interesting topic, the review protocol has been well written.

Introduction – has multiple ‘and’ in introduction, kindly consider reducing the number of ‘and’ in the introduction. Please update the references in the introduction. "why" you like to do this study needs to be elaborated and justified for better clarity

Review questions:

Question 2 to be reframed – not very clear

It might be good to mention the outcome measures you are looking at for each domain of outcomes like Well-being

QOL, Depression, Anxiety, stress to avoid confusions later during the review stage, as these domains are measured through multiple ways and means.

Methods:

Population: is very ambiguous, kindly consider narrowing it to a specific population.

Some detail on the setting and person delivering the intervention would be helpful. Will you consider studies that have used various healthcare professionals and also lay people performing the interventions - will that not be a bias on the intervention effectiveness ?

Looks like a scoping review might be a better option to perform considering your wide nature of research questions and scope of your research.

Appendix 1 - PRISMA P checklist - it might be good to add the page number or line number of the exact item where its present in the manuscript for easy identification. instead of just marking with "X"

Search strategy – Authors should detail the exact search terms and operators used for at least one database in the main manuscript, with the remaining details available in supplementary materials.

Minor: Font color to be kept uniform across the manuscript

Reviewer #2: General Comments: Incredibly interesting work and I think there is a value add for the broadening of topics that the author is suggesting. Overall, I think there is value to this systematic review that the author is proposing in the protocol. However, I believe this protocol paper is in need of several revisions in order to be published.

Abstract: Savouring should be introduced better. Why do we care about savouring? Why are savouring interventions important? I would recommend using potentially examples of when savouring has been beneficial. You mention in your introduction about previous systematic reviews. I would recommend adding clarification in your abstract on the gap that you are specifically filling. The goal of your review may be updated to include the extent of the literature on this topic and the overall evidence to support savouring interventions. If you are only including RCTs, why would you use grey literature? Any respectable RCT should be published in standard publishing channels: academic journals, peer-reviewed journals, etc. I have never seen a cutoff of 200. Particularly with a review that is hoping to broaden the scope of a topic; 200 seems limiting in my opinion. If you would like to keep in 200, I suggest adding a reference for this cutoff or some other explanation. If you are excluding papers that do not mention well-being, quality of life, depression, anxiety, or stress as a primary outcome, you need to explicitly state that. Add clarification about the reviewers and their roles. Please note some of these comments extend to other sections.

Introduction: I would add a paragraph about the use of savouring in interventions. Specifically, give the readers a tangible story of how savouring has been used and its effect.

Methods: I recommend removing “…for systematic reviews and meta-analyses…” You mention the PICO framework, yet your review questions do not include all of those specifics. Your questions should be more explicit. Also, I believe you are wanting “or stress.” If you mean “and stress” this would suggest that each study would need to measure all of these outcomes to be included. Again, I recommend removing grey literature. Adults must have some sort of diagnosed condition? Or could a study on healthy adults that examines savouring on self-reported stress levels be included? You say randomization is a fundamental inclusion criterion, yet you state studies with no intervention control will be included. Please clarify. Outcome – clarify if you mean “or stress” or “and stress.” Is the screening tool published anywhere? I suggest adding this tool to your supplemental materials if this is not already there.

Overall thoughts: There are several places in which clarification is required. As of now, it would be difficult to replicate this study. There is no mention of dates exclusions or word searches. If you are not excluding by date, this still should be mentioned. If you have a list of search terms, I recommend including your list. If you do not have your search terms finalized, I suggest developing this list. Your protocol should be replicable. Once significant clarification and detail has been added to the methods section, I believe this will be a worthwhile publication and eventual review. Good luck with the systematic review!

7. PLOS authors have the option to publish the peer review history of their article (what does this mean?). If published, this will include your full peer review and any attached files.

Reviewer #1: No

Reviewer #2: No

---

## [Author Response · Author response to Decision Letter 0]

26 Feb 2024

PLOS ONE Journal Requirements:

Comment 1: When submitting your revision, we need you to address these additional requirements.

and 

Response to Comment 1: These documents have been reviewed and we believe our manuscript now meets PLOS ONE’s styling requirements. 

Comment 2: We noticed you have some minor occurrence of overlapping text with the following previous publication(s), which needs to be addressed:

- https://doi.org/10.1080/17439760.2020.1818807

- https://bmcpsychiatry.biomedcentral.com/articles/10.1186/s12888-018-1739-2

In your revision ensure you cite all your sources (including your own works), and quote or rephrase any duplicated text outside the methods section. Further consideration is dependent on these concerns being addressed.

Response to Comment 2: We apologise for any overlapping text and can assure you this was done inadvertently. We believe our revised manuscript now correctly cites all referenced work. 

Additional Editor Comments:

1. It would be better to have two subheadings under the Methods section, e.g. “Identification and selection of studies” and “Assessment of characteristics of studies”.

Response: These subheadings have been included under the Methods section. 

2. Inclusion criteria should be spelled out as Design, Participants (not clear), Intervention (not clear, need to tidy up), Comparison (if applicable), Outcome measures.

Response: Inclusion criteria has now been restructured under these headings.

3. Please provide the exact search terms and operators used for at least one database in the main manuscript, with the remaining details available in supplementary materials. For each database, it should be as example like: Embase (1947 to June 2023). This is missing.

Response: Search terms and operators used for one database have been included within the main manuscript, with the remaining details available in supplementary materials. 

4. Authors should tidy up information, perhaps using a table to cover all area, such as Design, Participants, Intervention, Outcome measures, Comparisons.

Response: A table has been included to tidy up information outlined in written text. 

Main Editor Comments

1. Does the manuscript provide a valid rationale for the proposed study, with clearly identified and justified research questions?

Reviewer #1: Yes

Reviewer #2: Partly

Rationale for the proposed study has been expanded within the Introduction section of the protocol. The research questions have also been reframed to be more explicit. 

2. Is the protocol technically sound and planned in a manner that will lead to a meaningful outcome and allow testing the stated hypotheses?

Reviewer #1: Yes

Reviewer #2: Partly

The protocol has been revised to include further detail on the aims of the systematic review. Development has been made to better describe participants, interventions and outcomes being evaluated. The review questions have also been made more explicit. 

3. Is the methodology feasible and described in sufficient detail to allow the work to be replicable?

Reviewer #1: Yes

Reviewer #2: No

Additional detail has been added to the methodology section of the protocol. For example, an example of a search strategy for one database has been added to the main manuscript, with search strategies for all databases included as supplementary materials. Dates of database searches have also been made explicit. 

4. Have the authors described where all data underlying the findings will be made available when the study is complete?

Reviewer #1: No

Reviewer #2: No

Response: The following declaration has been included regarding data availability. 

Data Availability: Citations and full texts of studies included in the review will be fully available when the study is complete. Citations of papers excluded at full text screening will also be made available, sorted by reason for exclusion. All relevant data will be within the paper and its Supplementary Material files. 

5. Is the manuscript presented in an intelligible fashion and written in standard English?

Reviewer #1: Yes

Reviewer #2: Yes

6. Review Comments to the Author

Reviewer #1: 

Summary:

The authors have attempted to highlight the importance of savouring interventions on well being in adult clinical populations. Their main objective is to synthesise the literature and report a narrative synthesis with comparisons across the past, present and future focused savouring interventions.

Reviewer 1, Comment 1. Abstract: Kindly rephrase “no systematic review to date has been published evaluating their efficacy with adult clinical populations.” This is misleading as there are SRs evaluating their efficacy in adult populations. 

Response to Reviewer 1, Comment 1.

This phrase has been removed. The ‘Purpose’ section within the abstract has been rephrased: “Savouring interventions aim to amplify the intensity and duration of positive feelings and positive affect. Research has shown that the potential benefits of savouring include the promotion of psychological well-being and diminution of negative affective states. Savouring strategies may be particularly useful amongst clinical populations in changing biobehavioural processes which can strengthen an individual’s propensity to exert control over how to develop, intensify and promote psychological well-being, while simultaneously mitigating negative affective states. This paper outlines a protocol for a systematic review that will be used to identify, critically appraise and synthesise findings of studies examining the effectiveness of savouring interventions in adult clinical populations. Savouring interventions will be defined broadly, operationalised as any savouring strategy focusing on past, present or future events or experiences whereby participants are instructed to attend to and amplify positive affect relating to same. The goal of our review is to include the extent of the literature on this topic and contribute to the overall evidence to support savouring interventions.”

Reviewer 1, Comment 2. This is an interesting topic, the review protocol has been well written.

Introduction – has multiple ‘and’ in introduction, kindly consider reducing the number of ‘and’ in the introduction. 

Response to Reviewer 1, Comment 2.

This has been revised and edited. Six sentences including “and” from the introduction have been paraphrased:

1. “It has been proposed as a regulatory process which influences the relationship between positive events and plus an individual’s positive emotional reactions to these events [2]. 

2. This term was coined with a view of capturing the active process by which individuals perceive and experience enjoyment, and referencing an ongoing interplay between a person and their environment [3]. 

3. Savouring may be classified as a positive psychology intervention, as it pertains to processes that aim to regulate, intensify and as well as strengthen positive emotions and experiences.

4. Research has shown that the potential benefits of savouring include the promotion of psychological well-being and in addition to a decrease in experiencing negative affective states, most notably within depressive psychopathology [6, 7].

5. This points to the need to examine the efficacy of savouring interventions when applied to clinical populations on a range of clinical outcomes, including the strengthening of positive outcomes and together with the reduction of psychological distress, which may be bi-directional in nature.

6. Systematic reviews have previously been published which highlight the efficacy of particular savouring processes in improving participant levels of well-being and in addition to reducing symptoms of psychological distress, for example when practising gratitude [22] or when imagining one’s “Best Possible Self” [23].

Reviewer 1, Comment 3. Please update the references in the introduction. 

Response to Reviewer 1, Comment 3. 

References in the introduction have been updated. Furthermore, 13 additional references, cited within the introduction section of the manuscript have been added to the reference list:

1. Jain M, Pradhan M, Kar SK, Kumar P. Positive Psychology Interventions in Patients with Depression: A Review. Mind and Society. 2023 Jul 20;12(02):50-7.

2. Lauzon A, Green-Demers I. More of a good thing is even better: Towards a new conceptualization of the nature of savouring experiences. Journal of Happiness Studies. 2020 Apr;21(4):1225-49.

3. D'raven LL, Pasha-Zaidi N. Positive psychology interventions: A review for counselling practitioners. Canadian Journal of Counselling and Psychotherapy. 2014 Aug 12;48(4).

4. Pereira AS, Azhari A, Hong CA, Gaskin GE, Borelli JL, Esposito G. Savouring as an intervention to decrease negative affect in anxious mothers of children with autism and neurotypical children. Brain sciences. 2021 May 16;11(5):652.

5. Lyubomirsky S, Layous K. How do simple positive activities increase well-being?. Current directions in psychological science. 2013 Feb;22(1):57-62.

6. Quoidbach J, Berry EV, Hansenne M, Mikolajczak M. Positive emotion regulation and well-being: Comparing the impact of eight savoring and dampening strategies. Personality and individual differences. 2010 Oct 1;49(5):368-73.

7. Kahrilas IJ, Smith JL, Silton RL, Bryant FB. Savoring the moment: A link between affectivity and depression. International Journal of Wellbeing. 2020;10(2).

8. Stanton K, Stasik-O’Brien SM, Ellickson-Larew S, Watson D. Positive affectivity: Specificity of its facet level relations with psychopathology. Cognitive Therapy and Research. 2016 Oct;40:593-605.

9. Watson D, Clark LA, Carey G. Positive and negative affectivity and their relation to anxiety and depressive disorders. Journal of abnormal psychology. 1988 Aug;97(3):346.

10. Hurley DB, Kwon P. Savoring helps most when you have little: Interaction between savoring the moment and uplifts on positive affect and satisfaction with life. Journal of Happiness Studies. 2013 Aug;14:1261-71.

11. Quoidbach J, Mikolajczak M, Gross JJ. Positive interventions: An emotion regulation perspective. Psychological bulletin. 2015 May;141(3):655.

12. Carl JR, Soskin DP, Kerns C, Barlow DH. Positive emotion regulation in emotional disorders: A theoretical review. Clinical psychology review. 2013 Apr 1;33(3):343-60.

13. Ng W, Ong KR. Using positive psychological interventions to improve well-being: are they effective across cultures, for clinical and non-clinical samples?. Journal of Contemporary Psychotherapy. 2022 Mar;52(1):45-53.

Reviewer 1, Comment 4."Why" you like to do this study needs to be elaborated and justified for better clarity.

Response to Reviewer 1, Comment 4.

To provide greater clarify on the justification for this study the following paragraph has been added to the Introduction section of the manuscript. 

“Individuals living with chronic mental health or life limiting physical health conditions may be more likely to experience lower levels of positive affect or transiently positive emotional experiences [28], which may further dispose them to future psychological distress [29, 30]. This indicates that an individual’s disposition to initially experience positive affect states, and secondly attend to and savour them, may be impacted by their psychological and physical levels of functioning and well-being [31, 32]. Consequently, there is a need to examine the efficacy of savouring interventions when applied exclusively to clinical populations, as their therapeutic potential on altering positive and negative affect states may operate through different pathways than when applied to healthy, non-clinical populations [33, 34]. The goal of our review is to include the extent of the literature on this topic and contribute to the overall evidence to support savouring interventions.”

The following references, cited in the text above, have been added to the reference list:

1. Kahrilas IJ, Smith JL, Silton RL, Bryant FB. Savoring the moment: A link between affectivity and depression. International Journal of Wellbeing. 2020;10(2).

2. Stanton K, Stasik-O’Brien SM, Ellickson-Larew S, Watson D. Positive affectivity: Specificity of its facet level relations with psychopathology. Cognitive Therapy and Research. 2016 Oct;40:593-605.

3. Watson D, Clark LA, Carey G. Positive and negative affectivity and their relation to anxiety and depressive disorders. Journal of abnormal psychology. 1988 Aug;97(3):346.

4. Hurley DB, Kwon P. Savoring helps most when you have little: Interaction between savoring the moment and uplifts on positive affect and satisfaction with life. Journal of Happiness Studies. 2013 Aug;14:1261-71.

5. Quoidbach J, Mikolajczak M, Gross JJ. Positive interventions: An emotion regulation perspective. Psychological bulletin. 2015 May;141(3):655.

6. Carl JR, Soskin DP, Kerns C, Barlow DH. Positive emotion regulation in emotional disorders: A theoretical review. Clinical psychology review. 2013 Apr 1;33(3):343-60.

7. Ng W, Ong KR. Using positive psychological interventions to improve well-being: are they effective across cultures, for clinical and non-clinical samples?. Journal of Contemporary Psychotherapy. 2022 Mar;52(1):45-53.

Reviewer 1, Comment 5. 

1. Review questions: Question 2 to be reframed – not very clear

2. It might be good to mention the outcome measures you are looking at for each domain of outcomes like Well-being QOL, Depression, Anxiety, stress to avoid confusions later during the review stage, as these domains are measured through multiple ways and means.

Response to Reviewer 1, Comment 5.

1. Question 2 has been reframed: “What is the methodological quality of the evidence base for the effects of these savouring interventions, based on items from the Cochrane Collaboration’s tool for assessing risk of bias and the Jadad scale?” 

2. Outcome measures have been made more explicit within the Inclusion Criteria section of the Methods section of the manuscript: “Outcome measures: papers must include a primary outcome measure of well-being (e.g. happiness, life satisfaction), quality of life, depression, anxiety or stress. Assessment of intervention and control group outcomes must be assessed by self-report, clinician or proxy administered and validated psychometric instruments. Outcome measures may include but are not limited to the following tools; the Life Satisfaction Index, the Quality of Life Scale, The Center for Epidemiological Studies Depression Scale, the General Anxiety Disorder Scale and the Perceived Stress Scale [43-47].

The following references, cited in the text above, have been added to the reference list:

1. Adams DL. Analysis of a life satisfaction index. Journal of gerontology. 1969 Oct 1;24(4):470-4.

2. Heinrichs DW, Hanlon TE, Carpenter Jr WT. The Quality of Life Scale: an instrument for rating the schizophrenic deficit syndrome. Schizophrenia bulletin. 1984 Jan 1;10(3):388-98.

3. Radloff LS. The CES-D scale: A self-report depression scale for research in the general population. Applied psychological measurement. 1977 Jun;1(3):385-401.

4. Spitzer RL, Kroenke K, Williams JB, Löwe B. A brief measure for assessing generalized anxiety disorder: the GAD-7. Archives of internal medicine. 2006 May 22;166(10):1092-7.

5. Cohen S, Kamarck T, Mermelstein R. A global measure of perceived stress. Journal of health and social behavior. 1983 Dec 1:385-96.

Reviewer 1, Comment 6. 

Methods: 

1. Population: is very ambiguous, kindly consider narrowing it to a specific population.

2. Some detail on the setting and person delivering the intervention would be helpful. Will you consider studies that have used various healthcare professionals and also lay people performing the interventions - will that not be a bias on the intervention effectiveness?

3. Looks like a scoping review might be a better option to perform considering your wide nature of research questions and scope of your research.

Response to Reviewer 1, Comment 6.

1. Population: at present, this review is focused on participants who present with clinical problems, including mental health or physical health difficulties. This decision has been made as a key aim of this review is to expand the knowledge base of savouring interventions beyond their efficacy when applied to healthy, non-clinical populations. This clarification has now been added to the inclusion criteria section of the methods section within the manuscript: “Participants: adults (≥ 18 years) of any gender, who present with clinical symptoms or disorders. Participant mental health or physical health diagnosis must be given by medical professionals and/or based on meeting significant cut off points on validated psychometric instruments.”

2. Setting and person delivering the intervention: at present, no limitations have been placed on the professional background of people facilitating interventions, the settings in which interventions are delivered, or the mode of delivery. While we do appreciate that a variation in settings and in individuals facilitating interventions may impact intervention effectiveness, it is our goal to synthesise findings from multiple primary papers and to discuss the implications of same when comparing and contrasting settings and facilitators. It is our view that there is not a huge body of primary papers examining savouring interventions when applied to clinical populations and thus, being overly exclusive in terms of settings or facilitators of interventions may result in a very small number of papers being included in the final review which may result in a poor synthesis of the literature. 

3. Scoping review: at present, our rationale for conducting a systematic review in lieu of a scoping review is that, although this a broad topic, we are aiming to focus the research pertaining to the effectiveness of savouring interventions exclusively onto clinical populations and specific outcome measures. The research questions have been narrowed following feedback received from yourself, the editor and other reviewer. We hope that you may agree that a systematic review is an appropriate method to apply to answer these revised research questions. 

Reviewer 1, Comment 7. Appendix 1 - PRISMA P checklist - it might be good to add the page number or line number of the exact item where its present in the manuscript for easy identification. instead of just marking with "X"

Response to Reviewer 1, Comment 7.

Page numbers have been added to the PRISMA P checklist indicating where checklist items are present within the manuscript. 

Reviewer 1, Comment 8. Search strategy – Authors should detail the exact search terms and operators used for at least one database in the main manuscript, with the remaining details available in supplementary materials.

Response to Reviewer 1, Comment 8.

Breakdown of search terms used for one database have been added to the main manuscript, with additional details now available in supplementary materials. “The different terms identified to be used in searches can be seen in Table 1, with search strategies, with specific terms used, including Boolean operators for each database available in supplementary materials.”

Table 1: PsycINFO Search Terms for Systematic Review

AB (savoring OR savouring OR “positive reminisc*” OR basking OR “positive life review” OR “behavioural expression” OR “memory building” OR “self-congratulation” OR “positive anticipation” OR “positive emotion regulation” OR appreciation OR “mindful attention” OR awe OR wonder OR luxuriat* OR bask* OR thanksgiving OR marvel* OR “counting blessings” OR “three good things”)

Reviewer 1, Comment 9. Minor: Font color to be kept uniform across the manuscript

Response to Reviewer 1, Comment 9. 

Font colour is now consistent across the manuscript. 

Reviewer #2: General Comments: 

Incredibly interesting work and I think there is a value add for the broadening of topics that the author is suggesting. Overall, I think there is value to this systematic review that the author is proposing in the protocol. However, I believe this protocol paper is in need of several revisions in order to be published.

Reviewer 2, Comment 1. Abstract: Savouring should be introduced better. Why do we care about savouring? Why are savouring interventions important? I would recommend using potentially examples of when savouring has been beneficial. You mention in your introduction about previous systematic reviews. I would recommend adding clarification in your abstract on the gap that you are specifically filling. The goal of your review may be updated to include the extent of the literature on this topic and the overall evidence to support savouring interventions. 

Response to Reviewer 2, Comment 1.

Thank you for this feedback. The ‘purpose’ section of the Abstract has now been rephrased as a whole to expand on the utility and importance of examining savouring interventions and adding to the literature on same. 

“Savouring interventions aim to amplify the intensity and duration of positive feelings and positive affect. Research has shown that the potential benefits of savouring include the promotion of psychological well-being and diminution of negative affective states. Savouring strategies may be particularly useful amongst clinical populations in changing biobehavioural processes which can strengthen an individual’s propensity to exert control over how to develop, intensify and promote psychological well-being, while simultaneously mitigating negative affective states. This paper outlines a protocol for a systematic review that will be used to identify, critically appraise and synthesise findings of studies examining the effectiveness of savouring interventions in adult clinical populations. Savouring interventions will be defined broadly, operationalised as any savouring strategy focusing on past, present or future events or experiences whereby participants are instructed to attend to and amplify positive affect relating to same. The goal of our review is to include the extent of the literature on this topic and contribute to the overall evidence to support savouring interventions.”

Reviewer 2, Comment 2: If you are only including RCTs, why would you use grey literature? Any respectable RCT should be published in standard publishing channels: academic journals, peer-reviewed journals, etc. I have never seen a cutoff of 200. Particularly with a review that is hoping to broaden the scope of a topic; 200 seems limiting in my opinion. If you would like to keep in 200, I suggest adding a reference for this cutoff or some other explanation. 

Response to Reviewer 2, Comment 2.

1. Thank you for your comment regarding the inclusion of grey literature. 

Our rationale for including grey literature was underpinned by our decision to taking an inclusive approach to this systematic review. A rapid review has been conducted in which RCTs evaluating savouring interventions which meet our reviews inclusion criteria have been found which are not formally published, e.g. unpublished theses. It is our view that including grey literature may add to the methodological quality of the review and may have important implications for the reviews findings (Paez, 2017). However, we appreciate you may prefer if we remove grey literature from this review and would be happy to do so. 

2. Thank you for your comment regarding a cutoff of 200. 

For clarification, this cutoff number was imposed solely for results found from our Google Scholar search. This decision was made following guidance proposed by Haddaway et al., (2015) to focus on the first 200 results from a Google Scholar search when reviewed in conjunction with results reported by other electronic databases. This reference has now been added to the revised manuscript:

“The first 200 results, sorted by relevance, following a search on Google Scholar will also be searched for primary studies meeting the inclusion criteria outlined above [49].”

Reviewer 2, Comment 3: If you are excluding papers that do not mention well-being, quality of life, depression, anxiety, or stress as a primary outcome, you need to explicitly state that. 

Response to Reviewer 2, Comment 3.

This has been added to the Exclusion Criteria section of the method section: “RCTs of savouring interventions which do not include an outcome of well-being, quality of life, depression, anxiety or stress will be excluded.”

Reviewer 2, Comment 4: Add clarification about the reviewers and their roles. 

Response to Reviewer 2, Comment 4.

Clarification has been added within “Inter-Rater Agreement and Reliability of Coding” section: “KC, MM and ZBD will contribute to screening articles at both title and abstract and full text stages. Decisions of inclusion and exclusion of primary studies will be noted by individual raters, and disagreements resolved through discussion until a consensus is reached. In the event a consensus cannot be reached, consultation with FB will occur.”

Reviewer 2, Comment 5: Please note some of these comments extend to other sections.

Response to Reviewer 2, Comment 5.

Reviewer 2, Comment 6: Introduction: I would add a paragraph about the use of savouring in interventions. Specifically, give the readers a tangible story of how savouring has been used and its effect.

Response to Reviewer 2, Comment 6.

Further detail on the use of savouring in interventions has been added to the Introduction section of the manuscript:

“Savouring strategies have been implemented into interventions within the positive psychology literature in numerous ways. Examples of savouring strategies proposed by Bryant and Veroff [3] include sharing positive experiences with others, positive memory building, behavioural expression (laughing or showing affect), counting blessings, self-congratulation (regulating pride), engaging in positive imagination or fantasizing (e.g. attaining goals, imagining one’s “Best Possible Self”), reminiscing on past positive events and engaging in active-constructive communication with others [5]. Personal savouring interventions encourage an individual to reflect on personal positive events, memories or experiences, whereas relational savouring interventions consider positive moments one has had with other individuals, such as children, spouses, or other prominent individuals [8] Other savouring interventions include engaging in intentional positive activities such as practising kindness and expressing gratitude [9]. Interventions which include engagement with savouring strategies encourage the process of reliving, anticipating or attending to positive feelings associated with an experience or an event, whether real or imagined [10, 11]. Research has shown that the potential benefits of savouring include the promotion of psychological well-being in addition to a decrease in experiencing negative affective states, most notably within depressive psychopathology [6, 7]. Additionally, savouring processes can contribute to the development and maintenance of positive relationships with others, impact positively on mental and physical health outcomes, and can contribute to psychological agency as one ages [4, 8].”

Reviewer 2, Comment 7: Methods: I recommend removing “…for systematic reviews and meta-analyses…” 

Response to Reviewer 2, Comment 7.

This has been removed from the method section. 

Reviewer 2, Comment 8: You mention the PICO framework, yet your review questions do not include all of those specifics. Your questions should be more explicit. 

Response to Reviewer 2, Comment 8.

Review question 1 has been rephrased to be more explicit: “How effective are savouring interventions when applied to adult clinical samples on well-being, quality of life, depression, anxiety or stress, when compared with comparison control groups?”

Reviewer 2, Comment 9: Also, I believe you are wanting “or stress.” If you mean “and stress” this would suggest that each study would need to measure all of these outcomes to be included. 

Response to Reviewer 2, Comment 9.

Thank you for this correction. All references to “and stress” have now been corrected to “or stress”, noted in red text throughout the revised manuscript with tracked changes. 

Reviewer 2, Comment 10: Again, I recommend removing grey literature. 

Response to Reviewer 2, Comment 10.

Rationale for including grey literature has been addressed in Response to Reviewer 1, Comment 2 above. However, we appreciate you may still recommend removing grey literature from this review and would be happy to do so.

Reviewer 2, Comment 11: Adults must have some sort of diagnosed condition? Or could a study on healthy adults that examines savouring on self-reported stress levels be included? 

Response to Reviewer 2, Comment 11.

This has been clarified within the Inclusion Criteria section of the methods section: “Participants: adults (≥ 18 years) of any gender, who present with clinical symptoms or disorders. Participant mental health or physical health diagnosis must be given by medical professionals and/or based on meeting significant cut off points on validated psychometric instruments” and within the Exclusion Criteria section of the methods section: “Papers examining the effectiveness of savouring interventions on healthy adults who include outcomes such as well-being, quality of life, depression, anxiety or stress will not be included.”

Reviewer 2, Comment 12: You say randomization is a fundamental inclusion criterion, yet you state studies with no intervention control will be included. Please clarify. 

Response to Reviewer 2, Comment 12.

This has been addressed within the Inclusion Criteria section of the methods section. The phrase “no intervention control group” has been changed to “waiting list control group” to better explain no treatment or waiting list control groups. “Comparisons: studies with a waiting control group or alternative intervention control groups (e.g. treatment as usual, placebo control groups, or another well-defined treatment such as cognitive behavioural therapy) will be included.”

Reviewer 2, Comment 13: Outcome – clarify if you mean “or stress” or “and stress.” 

Response to Reviewer 2, Comment 13.

All references to “and stress” have now been corrected to “or stress”, noted in red text throughout the revised manuscript with tracked changes.

Reviewer 2, Comment 14: Is the screening tool published anywhere? I suggest adding this tool to your supplemental materials if this is not already there.

Response to Reviewer 2, Comment 14.

The screening tool has been added to supplementary materials. “KC, MM, ZBD and FB contributed to the development of a screening tool outlining the reviews inclusion and exclusion criteria, which is available in supplementary materials.”

Reviewer 2, Comment 15: Overall thoughts: There are several places in which clarification is required. As of now, it would be difficult to replicate this study. There is no mention of dates exclusions or word searches. If you are not excluding by date, this still should be mentioned. If you have a list of search terms, I recommend including your list. If you do not have your search terms finalized, I suggest developing this list. Your protocol should be replicable. Once significant clarification and detail has been added to the methods section, I believe this will be a worthwhile publication and eventual review. Good luck with the systematic review!

Response to Reviewer 2, Comment 15.

1. Search strategies

An example of the search strategy for one database is now included within the manuscript, with search strategies for all other databases now included in supplementary materials: 

The different terms identified to be used in searches can be seen in Table 1, with search strategies, with specific terms used, including Boolean operators for each database available in supplementary materials.

Table 1: PsycINFO Search Terms for Systematic Review

AB (savoring OR savouring OR “positive reminisc*” OR basking OR “positive life review” OR “behavioural expression” OR “memory building” OR “self-congratulation” OR “positive anticipation” OR “positive emotion regulation” OR appreciation OR “mindful attention” OR awe OR wonder OR luxuriat* OR bask* OR thanksgiving OR marvel* OR “counting blessings” OR “three good things”)

2. Dates exclusions

Dates of database searches have been clarified:

“Searches will be performed on databases from inception to present. No limitations will be imposed on date of publication.”

Additional Note:

Please note a final edit made to the revised manuscript to “The Status and Timeline of the Study” section. “Six months”, has been altered to “three months”, reflecting the updated status of the ongoing review.

“The review is ongoing. We expect to complete it and report results in three months, by April 2024.”

---

## [Decision Letter · Decision Letter 1]

26 Mar 2024

The effectiveness of savouring interventions on well-being in adult clinical populations: A protocol for a systematic review

PONE-D-23-36445R1

Dear Dr. Cullen,

We’re pleased to inform you that your manuscript has been judged scientifically suitable for publication and will be formally accepted for publication once it meets all outstanding technical requirements.

Kind regards,

Qin Xiang Ng, MBBS, MPH

Academic Editor

PLOS ONE

Additional Editor Comments (optional):

Reviewers' comments:

Reviewer's Responses to Questions

**Comments to the Author**

1. Does the manuscript provide a valid rationale for the proposed study, with clearly identified and justified research questions?

Reviewer #1: Yes

Reviewer #2: Yes

2. Is the protocol technically sound and planned in a manner that will lead to a meaningful outcome and allow testing the stated hypotheses?

Reviewer #1: Yes

Reviewer #2: Yes

3. Is the methodology feasible and described in sufficient detail to allow the work to be replicable?

Reviewer #1: Yes

Reviewer #2: Yes

4. Have the authors described where all data underlying the findings will be made available when the study is complete?

Reviewer #1: Yes

Reviewer #2: No

5. Is the manuscript presented in an intelligible fashion and written in standard English?

Reviewer #1: Yes

Reviewer #2: Yes

6. Review Comments to the Author

You may also provide optional suggestions and comments to authors that they might find helpful in planning their study.

Reviewer #1: Great job by the team to have addressed the questions! All the best for the completion of the review in April 2024.

I hope they find the needed articles for addressing their review questions.

Best Regards.

Reviewer #2: Thank you for your attention to detail in responding to my comments. In future publications on this work, be sure to give clear rationale for including grey literature. I understand your reasoning now, but it was unclear in your first draft.

7. PLOS authors have the option to publish the peer review history of their article (what does this mean?). If published, this will include your full peer review and any attached files.

Reviewer #1: No

Reviewer #2: No

---

## [Editor Report · Acceptance letter]

29 Mar 2024

PONE-D-23-36445R1 

PLOS ONE

Dear Dr. Cullen, 

I'm pleased to inform you that your manuscript has been deemed suitable for publication in PLOS ONE. Congratulations! Your manuscript is now being handed over to our production team.

Kind regards, 

on behalf of

Dr. Qin Xiang Ng 

Academic Editor

PLOS ONE